# An Annotated Inventory of Tanzanian Medicinal Plants Traditionally Used for the Treatment of Respiratory Bacterial Infections

**DOI:** 10.3390/plants11070931

**Published:** 2022-03-30

**Authors:** Ester Innocent, Alphonce Ignace Marealle, Peter Imming, Lucie Moeller

**Affiliations:** 1Department of Biological and Pre-Clinical Studies, Institute of Traditional Medicine, Muhimbili University of Health and Allied Sciences, United Nations Road, Dar es Salaam P.O. Box 65001, Tanzania; einnocent@muhas.ac.tz (E.I.); marealle2010@gmail.com (A.I.M.); 2Department of Clinical Pharmacy and Pharmacology, School of Pharmacy, Muhimbili University of Health and Allied Sciences, United Nations Road, Dar es Salaam P.O. Box 65013, Tanzania; 3Institute of Pharmacy, Faculty of Natural Sciences I, Martin Luther University Halle Wittenberg, Kurt-Mothes-Strasse 3, 06120 Halle (Saale), Germany; peter.imming@pharmazie.uni-halle.de; 4Department Centre for Environmental Biotechnology, Helmholtz Centre for Environmental Research–UFZ GmbH, Permoserstr. 15, 04318 Leipzig, Germany

**Keywords:** anti-infective plants, respiratory diseases, natural products, African traditional medicine, Tanzania

## Abstract

This review comprehensively covers and analyzes scientific information on plants used in Tanzanian traditional medicine against respiratory diseases. It covers ethnobotanical and ethnopharmacological information extracted from SciFinder, Google Scholar, and Reaxys as well as the literature collected at the Institute of Traditional Medicine in Dar-es-Salaam. Crude extracts and fractions of 133 plant species have literature reports on antimicrobial bioassays. Of these, 16 plant species had a minimum inhibitory activity of MIC ≤ 50 µg/mL. Structurally diverse compounds were reported for 49 plant species, of which 7 had constituents with MIC ≤ 5 µg/mL against various bacteria: *Bryophyllum pinnatum* (Lam.) Oken, *Warburgia ugandensis* Sprague, *Diospyros mespiliformis* Hochst. ex DC., *Cassia abbreviata* Oliv., *Entada abyssinica* A. Rich., *Strychnos spinosa* Lam., and *Milicia excelsa* (Welw.) C.C. Berg. The low number of antimicrobial active extracts and compounds suggests that antibacterial and antimycobacterial drug discovery needs to have a fresh look at ethnobotanical information, diverting from too reductionist an approach and better taking into account that the descriptions of symptoms and concepts of underlying diseases are different in traditional African and modern Western medicine. Nevertheless, some structurally diverse compounds found in anti-infective plants are highlighted in this review as worthy of detailed study and chemical modification.

## 1. Introduction

The United Republic of Tanzania lies between the latitudes 1° S and 12° S and longitudes 30° E and 40° E. Community health life of the people of this country is managed through established cultural norms and political administration. The government established a health system at all levels including dispensaries in villages, health facilities in wards, district and regional hospitals, zonal referral hospitals, and specialized and super-specialized national referral hospitals. Despite these well-established formal health facilities, community health-seeking behavior establishes that about 60% of the population consult traditional health practitioners as the first contact for their primary health care service [1]. Individuals often use the established community-based medicinal plants or those traded by vendors in herbal clinics or in open local markets [2].

Research on Tanzanian medicinal plants has been carried out by different research groups, especially from the Institute of Traditional Medicine and the School of Pharmacy of the Muhimbili University of Health and Allied Sciences; the Department of Traditional Medicine Research of the National Institute for Medical Research; the Departments of Microbiology and Biotechnology, Zoology, Botany, and Chemistry of the University of Dar-es-Salaam; the Departments of Veterinary Medicine and Public Health, and Chemistry and Physics, and the College of Forestry, Wildlife and Tourism of the Sokoine University of Agriculture; the Tanzania Tropical Pesticide Research Institute; and the School of Life Sciences and Bioengineering at the Nelson Mandela Institute of Science and Technology. Several studies have also been conducted by researchers residing outside Tanzania concerning the medicinal plants growing in the country.

This review gathers, consolidates, and clusters information on medicinal plant species to help researchers further integrate them in their scientific studies aiming at searching for antibacterial drug candidates for respiratory problems. This is motivated by the increasing number of reports on multi-resistant bacterial and mycobacterial strains to most of the drugs that are in the market [3]. Efforts to obtain new chemical entities from natural resources should not be underestimated. The current review includes plants that have been shown to inhibit bacteria that cause respiratory infections. Antifungal and antiviral plants are not part of this review.

Bacterial infections are one of the leading global burdens of infectious diseases, with a prevalence of about 18% airborne and 15% waterborne infections annually, the trend in Tanzania being the same [4]. Respiratory diseases constitute 5 of the 30 most common causes of diseases: chronic obstructive pulmonary diseases ranking 3rd, lower respiratory tract infections 4th, and tracheal, bronchial, and lung cancer 6th, while tuberculosis (TB) and asthma rank 12th and 28th [5]. Respiratory disease problems in Tanzania are exacerbated by asthma and chronic bronchitis associated with exposure to air pollutants [4], pneumonia, TB, and meningococcal meningitis. TB affects males twice as much as women in the age group 25–54 years [6]. Despite these threats, there is no established mechanism for data collection and monitoring of bacterial resistance to available drugs except for TB. Only occasionally are antimicrobial resistance situational analyses conducted. In Tanzania, an antimicrobial situational analysis in 2015 indicated that the resistance of *Streptococcus pneumoniae* to trimethoprim and sulphamethoxazole in children younger than five years increased from 25% in 2006 to 80% in 2012. This finding led to changes in the protocols for acute respiratory infection (ARI) treatment in children [3]. Several bacteria such as *Escherichia coli*, *Acinetobacter pyogenes*, *Staphylococcus hyicus*, *Staphylococcus intermedius*, and *Staphylococcus aureus* were resistant to antibiotics including ampicillin, benzylpenicillin, chloramphenicol, streptomycin, oxytetracycline, amoxicillin-clavulanate, sulphamethoxazole, and neomycin [3,7]. This situation indicates that the control and elimination of bacterial infections in the population have increasingly fewer drug options. The widespread occurrence of these infectious pathogens raises the need for searching for new, cheap, effective, and safe drugs from medicinal plants alongside the ongoing national programs of curbing antimicrobial resistance.

*Mycobacterium tuberculosis* is mainly transmitted via infectious droplets that are spread by infected individuals through coughing and sneezing. Patients carrying many bacilli are the main source of infection, while other people may be carriers without developing the disease nor posing a significant risk of transmission and infection [8]. The World Health Organization (WHO) for the year 2016 reported more than 10 million cases of active TB worldwide, leading to an estimated 1.3 million deaths and TB being the number one cause of death among all infectious diseases before COVID-19 [6]. Knowing the threats of mycobacterial infections in the population of Tanzania, the National Tuberculosis and Leprosy Programme, which was established in 1977, performs a routine TB surveillance with incidence as an indicator for the disease burden. The program involves invited participants in the survey or enrolled identified TB suspects. The participants are screened by a simple symptom questionnaire, a chest X-ray, and assessment of microscopy of sputum specimens [9,10]. This program makes Tanzania one of the countries using the standard Direct Observed Therapy Short (DOTS) Course to treat TB. Tanzania is also classified as 1 of the 22 countries with a high burden of TB in the world [6]. Consequently, the Tanzania National Tuberculosis Prevalence Survey of bacteriology reported TB affecting 200–299 per 100,000 people, mostly adults of the age group 54 years or older [6,11]. Another report showed a bigger burden for males compared to women of the age group between 25 and 54 years [6]. TB mortality in Tanzania was 5.8% in 2012, a decrease of 11.6% from that reported in 2003. Often, people suffering from TB have co-infections either with HIV [12], helminths [13], or malaria [14]. TB accounts for about 8% of the burden of diseases and 6% of all deaths for people aged 5 years and above, primarily due to HIV/TB co-infection [9]. The burden of disease is sometimes exacerbated by difficulties in the diagnosis of TB cases when there is an absence of cough, connected with an inability to produce sputum [8,9]. One of the alternative management avenues available for the indigenous rural communities in Tanzania is the use of available bioactive antimycobacterial agents from medicinal plants growing in the areas.

Plants possess a myriad of chemical compounds—secondary metabolites—which are partly produced for plant defense. Naturally, plants do not locomote, so they are susceptible to attack from the surrounding environment, including herbivorous animals, arthropods, fungi, and bacteria as well as human activities. The most vulnerable exposed parts for attacks are the roots, leaves, and stem barks, which are also parts researchers are interested in for searching for bioactive compounds. The role of these secondary bioactive compounds is to interact and intercept unfavorable infective agents that may harm the plant [15]. Some of these compounds happen to be effective against various Gram-positive and Gram-negative bacteria.

As yet, very few natural products originating from plants are qualified for drug development. Most of the natural products in clinical studies do not originate from plants [16]. Still, natural products are considered to be good sources of structurally new active compounds, as exemplified by pleuromutilin which was first isolated from the fungus *Clitopilus passeckerianus* and then from the insect *Drosophila subatrata* (drosophilin B) [17]. Pleuromutilin—biosynthetically derived from mevalonic acid—inhibits protein synthesis in bacteria and has potential in the treatment of infectious respiratory tract diseases.

## 2. Results

### 2.1. Ethnobotanical Information of Medicinal Plants against Infective Respiratory Tract Diseases

A total of 169 plant species from 54 families were identified to be used against diseases of the respiratory tract in Tanzanian traditional medicine (Appendix A). Of these, 93 plant species (equivalent to 55.0%) came from 9 families, with plant species in the families Leguminosae (Fabaceae) (27; 16.0%), Euphorbiaceae (11; 6.5%), Malvaceae (10; 5.9%), and Asteraceae (10; 5.9%) dominating. Others were Malvaceae (10; 5.9%), Bignoniaceae (8; 4.7%), Rutaceae (8; 4.7%), Rubiaceae (8; 4.7%), Phyllanthaceae (7; 4.15%), Anacardiaceae (7; 4.1%), and Lamiaceae (5; 3.0%). A similar trend of plant families has been reported [18,19,20]. The most cited plant in our review was *Zanthoxylum chalybeum* Engl. Six publications reported its widespread use against diseases such as TB, cough, and pneumonia as well as malaria and intestinal worms. *Hoslundia opposita* Vahl was cited five times. Regarding lung diseases, this plant is used against cough only.

Medicinal plant species were reported most often to be used for the alleviation of cough (99; 59.0%) and TB (66; 39%), followed by pneumonia (50; 30%). Not surprisingly, some plant species were used against one or several respiratory tract diseases. Cough was the common description of respiratory problems facing the communities. It was described varyingly as bloody cough, chronic cough, severe cough, persistent cough, whooping cough, bilious cough, etc., depending on the seriousness of chest pain and character of the sputum manifested.

The vast majority of plants were either trees or shrubs (133, i.e., 78.7%), while only 27 (15.4%) plant species were herbs. The most frequently used part of the plant was the root (101 times, i.e., 59.8%), followed by the leaves (61 times, i.e., 36.1%) and stem barks (48 times, i.e., 28.4%). Fruits and seeds, entire plants, and the aerial parts were used only rarely (Appendix A).

Forty per cent of the literature sources were older than twenty years; one third were younger than ten years. Seven papers were published by Chhabra et al., followed by three papers written by Hedberg as the first author. These two persons provided exhaustive general overviews on the ethnobotany of medicinal plants in Tanzania.

### 2.2. Ethnopharmacology of Extracts from Bioactive Plant Species against Infective Bacteria of Respiratory Tract

Appendix A lists extracts and fractions from 133 different plant species that were tested against diverse bacteria.

Reported plant extracts tested most often included *Kigelia africana* (Lam.) Benth. (13 publications), *Achyranthes aspera* L., *Dichrostachys cinerea* (L.) Wight & Arn., and *Securidaca longipedunculata* Fresen. (10 publications each). Extracts of these plants showed high antibacterial activity in at least one assay. Extracts from 69 plant species were tested for their anti-TB activities, 31 of which had been previously reported to be used in Tanzanian traditional medicine against TB (compare Appendix A). Review articles were written about three plants: *Acacia nilotica* (L.) Delile [21], *Peltophorum africanum* Sond. [22], and *Lantana camara* L. [23]. Therefore, no extensive literature search for these three plant species was performed for this review.

In most tests, extracts were subjected to an agar diffusion test to measure inhibition zone diameters (IZDs), and/or the minimum inhibitory concentration (MIC) was determined by broth microdilution methods. Due to the variety of assay details and conditions, the results cannot be reliably compared quantitatively to find the most active extracts. There were 22 extracts with MIC ≤ 50 µg/mL and 44 extracts with IZD ≥ 20 mm. Regarding mycobacteria, extracts and fractions of four plants showed good activities against *M. tuberculosis*, another four against *M. smegmatis*, and one against *M. aurum* and *M. phlei*. Regarding other bacteria causing respiratory diseases, 45 extracts were highly active against *Staphylococcus aureus*, 13 extracts and 1 essential oil showed activity against *Pseudomonas aeruginosa*, extracts from another 5 plant species showed activity against *Klebsiella pneumoniae*, and 1 extract showed activity against *Serratia marcescens*. The most active plant extracts against bacteria that cause respiratory diseases are listed in Table 1. The highest activity was shown by the ethanol leaf extract of *Syzygium cordatum* Hochst. ex Krauss, with MIC = 0.01 µg/mL, against *S. aureus* [24]. Dichloromethane and water leaf extracts of this plant also showed high activity against *K. pneumoniae*, with MIC = 0.39 µg/mL [24].

Two thirds of the literature cited in Appendix A was published between 2010 and 2019. One quarter was published between 2000 and 2010, and only 9% in the time before 2000. This shows the increasing intensity in the search for anti-infective drugs based on Tanzanian natural resources. 

### 2.3. Bioactivity of Chemical Compounds Isolated from Antibacterial Bioactive Plant Species

Appendix A lists 265 isolated bioactive compounds from 49 plant species. Of the 202 references cited in Appendix A, 68 (34%) were older than 10 years and only 17 (8%) were published before 2000. This reflects the steadily improving possibilities for the elucidation of molecular structures in Tanzania and worldwide.

All isolated compounds were tested for antimicrobial activity. A total of 68 (i.e., 26%) compounds from 31 plant species met the criteria presented in the previous chapter (MIC ≤ 50 µg/mL and IZD ≥ 20 mm). Seventeen compounds were very active, showing MIC ≤ 10 µg/mL or IZD ≥ 25 mm. We suggest a cut-off for MIC of 5 µg/mL for individual compounds meriting further investigation as often established in assays of antimycobacterial drugs [39,40]. Based on this, in the following paragraph, the compounds are highlighted that reportedly had such activity or better against bacteria.

Two anthraquinones isolated from *Rubia cordifolia* L. met well our thresholds (rubiacordone A: 29 ± 0.2 mm for *Streptococcus faecalis* and IZD = 26 ± 0.3 mm for *Bacillus subtilis*, and 1-acetoxy-6-hydroxy-2-methylanthraquinone-3-*O*-[α-L-rhamnopyranosyl-(1→2)-β-D-glucopyranoside]: IZD = 27 ± 0.3 mm for *B. subtilis* and 26 ± 0.5 mm for *Bacillus cereus*) [41], as did kaempferol 3-*O*-α-d-glucopyranoside-7-*O*-α-*L*-rhamnopyranoside (Figure 1A) (MIC = 1 µg/mL against *Salmonella typhi*, and MIC = 4 µg/mL against *S. aureus* and *P. aeruginosa*), *ɑ*-rhamnoisorobin (Figure 1B) (activity against *P. aeruginosa* and *S. typhi* with MIC = 1 µg/mL each, and *S. aureus* with MIC = 2 µg/mL), and afzelin (MIC = 4 µg/mL against *S. typhi*), all isolated from *Bryophyllum pinnatum* (Lam.) Oken [42], and xanthonoid mangiferin from *Mangiferia indica* L. (IZD = 26 ± 0.30 mm for *Salmonella agona* [43], and IZD = 29 mm for *Salmonella virchow* [44]. Linoleic acid from *Warburgia ugandensis* Sprague (Canellaceae) showed activity against *M. aurum* and *M. phlei*, with MIC values of 4 µg/mL [45]. *Diospyros mespiliformis* Hochst. ex DC. (Ebenaceae) contains a naphthoquinone epoxide diosquinone (Figure 1C) with activity against *S. aureus* NCTC 6571 (MIC = 3 µg/mL), *S. aureus* E3T, and *B. subtilis* (MIC values of 5 µg/mL each) [46]. The family Fabaceae afforded two plant species with highly active compounds. *Cassia abbreviata* Oliv. (Fabaceae) contains proanthocyanidin cassinidin A (Figure 1D), which had activity against *E. coli* (MIC = 1 µg/mL), while against *B. subtilis* and *S. aureus*, the MIC values were 0.5 µg/mL each. Another proanthocyanidin, cassinidin B, from the same plant species exhibited activity against *B. subtilis* and *S. aureus*, with MIC values of 5 µg/mL each [47]. *Entada abyssinica* A. Rich. (Fabaceae) contains quercitrin, a glycoside formed from the flavonoid quercetin and rhamnose, with an MIC of 3.12 µg/mL against *S. typhimurium* and entadanin (Figure 1E), and an MIC of 1.56 µg/mL against *S. typhimurium* [48]. From *Strychnos spinosa* Lam. (Loganiaceae), sarracenin (Figure 1F) was isolated and exhibited activity against *S. aureus*, *E. coli*, *S. dysenterae*, and *K. pneumoniae*, with MIC = 2.5 µg/mL each, while the activity against *S. pyogenes*, *S. typhi*, and *P. aeruginosa* had an MIC value of 5 µg/mL [49]. The flavonoid neocyclomorusin isolated from *Milicia excelsa* (Welw.) C.C. Berg (Moraceae) exhibited activity against *K. pneumoniae* ATCC11296 and *E. cloacae* BM47, with MIC values of 4 µg/mL each [50].

These rather diverse compounds should be further explored as antibacterial lead structures.

## 3. Discussion and Recommendation

The following lessons from the information gathered herein indicate at least three challenges that researchers in anti-infectives need to deal with when striving for new drug discoveries from plants. The first is to be reductionist in the research process by assuming that the activity of constituents increases with purity. The reason could be that sometimes the various constituents in extracts may act synergistically rather than individual entities that bioguided fractionation only yields. For example, the ethanolic and methanolic extracts of Psidium guajava L (Myristaceae) leaves exhibited higher activity compared to each individual isolated compound [37,51]. The same is true for the petroleum ether extract of Gymnosporia senegalensis (Lam.) Loes (syn. Maytenus senegalensis (Lam.) Excell.) (Celastraceae) [24,52,53] and that of Spirostachys africana Sond. (Euphorbiaceae) [24,54], a phenomenon observed before [55,56]. Similarly, Appendix A lists the ethnobotanical uses of 169 plant species from 54 plant families. Surprisingly, none of the ethnobotanically dominating plant families yielded a very active compound except Fabaceae. The fact that there are only a few antibacterial compounds from ethnobotanical plant collections may also underscore that there is plenty of indigenous knowledge or information about these plants, their preparation, and use that researchers fail to grasp when collecting ethnobotanical information [57] or are not able to reproduce such information using laboratory methods.

Another finding that emerges from this review is the small number of bioactive lead compounds found in plants to date. Appendix A lists 133 plant species whose extracts were tested for antimicrobial activity, with only 22 having MIC ≤ 50 µg/mL and 44 having IZD ≥ 20 mm. The reason could be that it is difficult to reproduce traditional healers’ claims because they usually use water extracts, while most reported antimicrobial compounds from plants are not polar [58]. Moreover, some discrepancies between field and lab findings may be due to factors that affect the growth of microbials such as the choice of test method, dilution of culture media, concentrated inoculum, and solvents that do not allow a comparison of the results [59]. As evidenced in Appendix A, some researchers combined at least two antimicrobial test methods starting with agar or disc methods followed by a broth dilution method either without or with indicators. Variations in these methods bring different scales of comparison for fair judgement among tested samples. Furthermore, agar is an aqueous preparation, while most researched plant compounds are non-polar; therefore, these compounds will not diffuse, leading to no activity or small inhibition zones [57,58].

Lastly, although many plants are ethnobotanically used in traditional medicine to relieve respiratory disorders, a conservation dilemma exists, especially for large-scale consumption. The use of roots of course endangers local plant populations or even whole species if the roots are harvested without desirable care, because the whole plant is at risk compared to the harvest of other parts such as the leaves, fruits, or stem bark. The dilemma in using roots exists because most bioactive compounds identified thus far are mainly or exclusively located in this part of the plant. This can be explained in terms of defense and survival mechanisms of plants. The roots are much more susceptible to attack by microbes and pests and hence contain the vast majority of secondary metabolites such as directly or indirectly antimicrobial phytoalexins and phytoanticipin [15,60,61].

Furthermore, many of the plants yielding antimicrobial medications—as our review and other studies [19] show—are trees or shrubs which generally grow more slowly than herbs, ferns, succulents, or lianas. For corroboration, a recent review by Alamgeer et al. reported 384 plant species belonging to 85 families used to treat respiratory disorders in Pakistan, with their habits mostly being herbs (219), shrubs (112), and trees (69) [62]. A similar trend has been reported [18,19,20] and agrees with what we found (Appendix A).

Therefore, our recommendation is to consider ethnobotanical information in the research processes in order to increase the probability of generating new microbial agents. The low number of antimicrobial active extracts and compounds suggests a fresh look diverting from too reductionist an approach and better considering that the descriptions of symptoms and concepts of underlying diseases are different in traditional African and modern Western medicine. Additionally, renewed efforts to explore identified bioactive chemical compounds from plants through structure–activity relationship studies, synthesis of analogues, and high-throughput screening for such active compounds and derivatives could increase the likelihood of obtaining new microbial agents from plants. 

## 4. Methodology Utilized in Collection of Information

This review covers the ethnobotanical and ethnopharmacological literature from 1982 to 2019. Information was gathered from SciFinder, Chemical Abstract, Pubmed, Springerlink, Science Direct, Scopus, the Web of Science, Google Scholar, and ResearchGate. Key search words used included cough, pneumonia, and tuberculosis, the leading symptoms of respiratory tract diseases. The Tanzanian coauthors’ direct access to local information and literature was complemented by a broad database search to confirm the pharmacological activity of extracts and single compounds. Antimicrobial tests against bacteria that cause respiratory diseases were primarily included and complemented by other reports on the antimicrobial activity of the plant species, extracts, and constituents.

## Figures and Tables

**Figure 1 plants-11-00931-f001:**
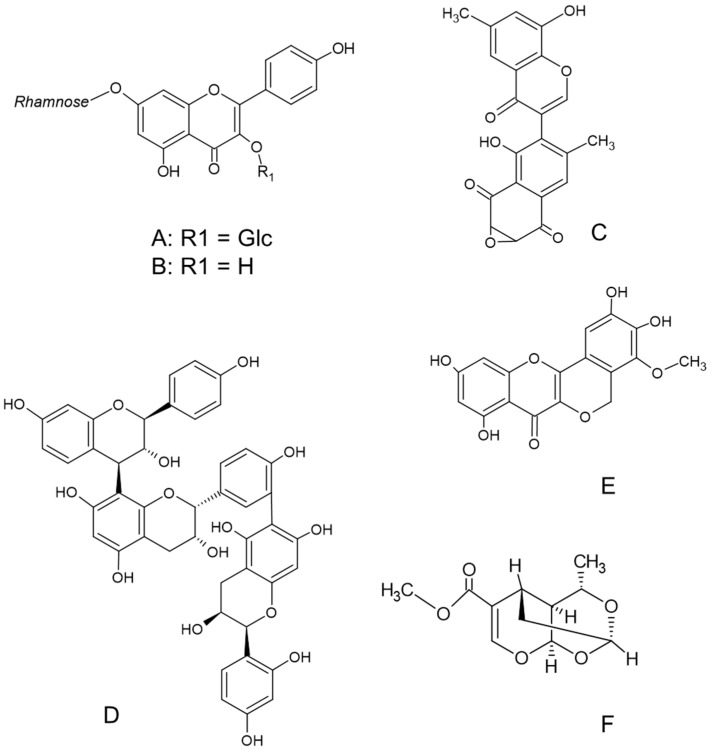
Structures of the most important compounds: kaempferol 3-*O*-α-D-glucopyranoside-7-*O*-α-l-rhamnopyranoside (**A**) [42], ɑ-rhamnoisorobin (**B**) [42], diosquinone (**C**) [46], cassinidin A (**D**) [47], etandanin (**E**) [48], and sarracenin (**F**) [49].

**Table 1 plants-11-00931-t001:** Most active plant extracts against bacteria causing respiratory diseases with MIC ≤ 50 µg/mL (note: only MIC was used for the selection of the plants for Table 1).

Bacteria	Plant Extract with Activity	Reference
*Klebsiella pneumoniae*	*Gymnosporia senegalensis* (Lam.) Loes (root petroleum ether extract, dichloromethane extract, and ethanol extract with MIC = 1.56 µg/mL each; water extract with MIC = 3.13 µg/mL)	[24]
	*Spirostachys africana* Sond. (bark petroleum ether, dichloromethane, and ethanol extracts with MIC = 1.56 µg/mL each)	[24]
	*Cassia abbreviata* Oliv. (stem bark trichloromethane extract with MIC = 46.88 µg/mL)	[25]
	*Syzygium cordatum* Hochst. ex Krauss (leaf petroleum ether extract with MIC = 6.25 µg/mL, dichloromethane and water extracts with MIC = 0.39 µg/mL each, ethanol extract with MIC = 1.56 µg/mL)	[24]
*Mycobacterium tuberculosis*	*Warburgia salutaris* (Bertol.f.) Chiov. (leaf acetone extract against MTB (H37Ra) with MIC = 25 ± 2 µg/mL and MTB 2 with MIC = 25 ± 5 µg/mL)	[26]
	*Bryophyllum pinnatum* (Lam.) Kurz. (leaf *n*-hexane fraction and dichloromethane fractions of methanol extract with MIC = 25 µg/mL both, ethyl acetate fraction of methanol extract and water extract with MIC = 40 µg/mL both)	[27]
	*Bridelia micrantha* (Hochst.) Baill. (bark acetone extract against MTB (H37Ra) with MIC = 25 ± 0 µg/mL and against MTB 2 with MIC = 25 ± 1 µg/mL)	[28]
	*Lantana camara* L. (leaf methanol extract against MTB (H37Rv) with MIC = 20 µg/mL and MTB (TMC-331) and wild strain with MIC= 15 µg/mL both)	[29]
*Pseudomonas aeruginosa*	*Ozoroa mucronata* (Bernh.) R. Fern & A. Fern. (leaf dichloromethane fraction of crude (acetone (70%)-*n*-hexane) extract dichloromethane fraction with MIC = 39 µg/mL)	[30]
	*Cassia abbreviata* Oliv. (stem bark cold water extract with MIC = 46.88 µg/mL)	[25]
	*Trichilia emetica* Vahl (leaf dichloromethane/methanol (1:1) extract with MIC = 30 µg/mL and seed dichloromethane extract with MIC = 31 µg/mL)	[31,32]
	*Phyllanthus amarus* Schumach. & Thonn. (leaf water extract against *P. aeruginosa* with MIC = 30 µg/mL and *P. aeruginosa* NCTC10662 with MIC = 30 µg/mL, and ethanol extract against *P. aerugionsa* with MIC = 30 µg/mL and *P. aeruginosa* NCTC10662 with MIC = 35 µg/mL)	[33]
	*Citrus limon* (L.) Osbeck (essential oil with MIC = 12.5 µg/mL)	[25]
*Staphylococcus aureus*	*Ozoroa mucronata* (Bernh.) R. Fern & A. Fern. (leaf crude (acetone (70%)-*n*-hexane) extract dichloromethane fraction with MIC = 19 µg/mL)	[30]
	*Elaeodendron buchananii* (Loes.) Loes. (stem bark ethyl acetate extract with MIC = 15.62 µg/mL)	[34]
	*Gymnosporia senegalensis* (Lam.) Loes (root petroleum ether extract with MIC = 6.25 µg/mL, dichloromethane and ethanol extracts with MIC = 0.78 µg/mL each)	[24]
	*Solanecio mannii* (Hook.f.) C. Jeffrey (leaf cyclohexane extract with MIC = 6.3 µg/mL)	[35]
	*Spirostachys africana* Sond. (bark petroleum ether extract with MIC = 0.39 µg/mL, dichloromethane extract with MIC = 3.13 µg/mL, ethanol extract with MIC = 0.01 µg/mL, and water extract with MIC = 0.78 µg/mL)	[24]
	*Cassia abbreviata* Oliv. (stem bark methanol extract with MIC = 15 µg/mL)	[31]
	*Tamarindus indica* L. (flower methanol extract with MIC = 25 µg/mL)	[36]
	*Psidium guajava* L. (leaf methanol extract with MIC = 25 µg/mL)	[37]
	*Syzygium cordatum* Hochst. ex Krauss (leaf petroleum ether extract with MIC = 6.25 µg/mL, dichloromethane extract with MIC = 0.20 µg/mL, ethanol extract with MIC = 0.01 µg/mL, and water extract with MIC = 0.78 µg/mL)	[24]
	*Phyllanthus amarus* Schumach. & Thonn. (leaf water extract with MIC = 20 µg/mL against *S.aureus* NCTC6571 and ethanol extract with MIC = 20 µg/mL against both *S. aureus* and *S. aureus* NCTC6571)	[33]
	*Citrus limon* (L.) Osbeck (essential oil with MIC = 12.5 µg/mL)	[38]

## Data Availability

Not applicable.

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
