# Peer review of "An Annotated Inventory of Tanzanian Medicinal Plants Traditionally Used for the Treatment of Respiratory Bacterial Infections"

_plants, 2022, doi:10.3390/plants11070931_

Round 1
Reviewer 1 Report
Please rewrite and submit
Author Response
We thank the reviewer for the time spent with our large draft and his evaluation. We again checked and improved the language.
Reviewer 2 Report
This review covers and analyses scientific information about plants used in Tanzanian traditional medicine against respiratory diseases, plants that have been shown to inhibit bacteria that cause respiratory infections. The prevalent manifestation of these infectious pathogens raises the need for searching new, cheap, effective, and safe drugs from medicinal plants, so this kind of article is interesting, because it allows to develop a register of these plants and their bioactive compounds.
The authors stated that forty per cent of the literature sources were older than twenty years and one-third was younger than ten years, however, there are some papers published in 2020 that were not cited in this article and can be important. For example:
Obakiro, S. B., Kiprop, A., Kowino, I., Kigondu, E., Odero, M. P., Omara, T., & Bunalema, L. (2020). Ethnobotany, ethnopharmacology, and phytochemistry of traditional medicinal plants used in the management of symptoms of tuberculosis in East Africa: a systematic review. Tropical Medicine and Health, 48(1), 1-21.
Sharifi-Rad, J., Salehi, B., Stojanović-Radić, Z. Z., Fokou, P. V. T., Sharifi-Rad, M., Mahady, G. B., ... & Iriti, M. (2020). Medicinal plants used in the treatment of tuberculosis-Ethnobotanical and ethnopharmacological approaches. Biotechnology advances, 44, 107629.
Otieno, J. N. (2020). Conservation Assessment of Plants Used for Respiratory Diseases by Using Ethnobotanical Criteria: Case of Lake Victoria Region, Tanzania. Recent Advances in Biological Research Vol. 6, 22-32.
It could be interesting, look for the difference between the plants found by Obakiro et al. 2020, Sharifi-Rad et al., 2020 and this work.
I suggest extending the review at least until 2020 and if it is possible to 2021
Author Response
We thank the reviewer for the time spent with our large draft, the positive appraisal and the 2020 references suggested which we included in the revision. Unfortunately, it is not possible for us to extend the review to literature from 2020 and 2021 due to the short follow-up time and the lack of manpower.
Reviewer 3 Report
I highly appreciate the submitted review article on ethnobotany, medicinal plants and their use in the treatment of diseases (the respiratory bacterial infections) in the African country. In Tanzania, traditional medicine provides health care and support to over 60 % of the rural population. This trend is mainly due to the strong attachment to traditions and spirituality and to the greater access, with respect to conventional medicine, to healers inside villages that provide low cost treatments. The traditional medical treatments are mainly based on herbal remedies, using sometimes many different species mixed together.
In regard to the methodology utilized in collection of information – this review covers the ethnobotanical and ethnopharmacological literature from 1982 108 to 2019. Information was gathered from SciFinder, Chemical Abstract, Pubmed, Spring-109 erlink, Science Direct, Scopus, the Web of Science, Google Scholar and Research gat. I appreciate the use of a large number of literary sources, of which there are more than 300, the systematic and hard scientific work has been done in this regard.
The summary paper shows high quality which I expect in a first-class international journal. All parts are written and modified according to instruction for authors for Journal of Plants MDPI. Abstract section consists all needed information. Innovation point of this study is high and authors gave sufficient credit to related work. Manuscript does not need language revision only minor revision from authors to correct few uncomplete points. Topic is relevant and the content of the manuscript is in line with policy of the journal. I recommend this paper for publication in Journal of Plants MDPI.
Author Response
We thank the reviewer for the time spent with our large draft and particularly for the encouraging appraisal of our work.
Reviewer 4 Report
The review by Ester Innocent and others provides a comprehensive and reasonably complete survey on the most important medicinal plants, known for Tanzania and used in Tanzania. The authors describe drugs that show effects at practically useful concentrations.
The review focusses on bacterial infection treatment in that region, including many bacteria that often give rise to serious complication including Mycobacterium tuberculosis and Staphylococcus aureus. Also in this respect, the manuscript provides many inspirations for future research on active plant substances.
Where appropriate and possible, the authors provide profound information on chemical properties of the active substances together with MIC values. Both render thois contribution highly useful for the community.
The literature cited, very important in a review, seems to be well-chosen nearly complete.
The manuscript is essentially well-written, easy to read and it sheds light on often underestimated procedures and drugs.
The reviewer has no doubt that this contribution is a valuable addition to knowledge on medicinal plants and recommends the manuscript for publication without any reservation.
Author Response
We thank the reviewer for the diligent perusal of our large draft. The comments were taken care of.
Reviewer 5 Report
Review is comprehensive and very well written. A typo error in line 125, venerable should be vulnerable. There are a few words with "-" in between, e.g. for-mal in line 41. Please correct.
Author Response

(The authors gave the same response as above.)

Round 2
Reviewer 1 Report
The authors improve this manuscript.
Introduction: Please discard the subtitle from the Introduction.
Each subtitle part makes a paragraph.
Result: Each subtitle should make concisely and meaningful. No needs always write reviewed ---.
Figure 1. The authors should use chem draw to make their structure (if possible) to reduce the number of unwanted citations (ref42, 46-49) and increase the possibility of their citation.
Tables should add to the main manuscript. If possible, please try to be concise.
If possible, please separate the Recommendation from Discussion and Recommendation.
In Discussion and Recommendation, the authors used subtitles. 3.1, 3.2, and 3.1. I do not find any relevant challenge writing. Why use the challenge word? Please check and improve this part carefully.
In methodology, the authors used 1982-2019 data. The authors should extend until 2021. If the authors use recent ten years' data, it would be excellent for readers.
The number of references is 309—too many. The authors should reduce references as many as possible from old and irrelevant.
Author Response
Dear Reviewer,
thank you for reading our manuscript. You can find our answers to your questions in the attached file.
Best regards
Lucie Moeller
